# Disentangling the effect of ocean temperatures and isotopic content on the oxygen - isotope signals in the North Atlantic Ocean during Heinrich Event 1 using a global climate model

Marianne Bügelmayer-Blaschek<sup>1</sup>, Didier M. Roche<sup>1,2</sup>, Hans Renssen<sup>1</sup>, and Claire Waelbroeck<sup>2</sup>

<sup>1</sup>Earth and Climate Cluster, Faculty of Earth and Life Sciences, Vrije Universiteit Amsterdam, Amsterdam, The Netherlands

<sup>2</sup>Laboratoire des Sciences du Climat et de l'Environnement (LSCE), CEA/CNRS-INSU/UVSQ, Gif-sur-Yvette Cedex, France

Correspondence to: M. Bügelmayer-Blaschek(m.bugelmayer@vu.nl)

Abstract. Heinrich events are intriguing episodes of enhanced iceberg discharge occurring during the last glacial period and are characterized by a steep increase in ice rafted debris (IRD) found in North Atlantic cores. Yet, their signal is not directly recognizable in the carbonate oxygen isotopic composition recorded in planktonic foraminifera, which depends on both the prevailing tempera-

- ture and isotopic composition of seawater. Using the global isotope-enabled climate-iceberg model *i*LOVECLIM we performed three experiments to shed light on first, the impact of the duration of a Heinrich event-like iceberg forcing on the North Atlantic Ocean and second, the mechanisms behind the simulated  $\delta^{18}O_{calcite}$  pattern. We applied an iceberg forcing of 0.2 Sv for 300, 600 and 900 years, respectively, and find a strong and non-linear response of the Atlantic Meridional Overturning
- Circulation (AMOC) to the duration of the Heinrich event in *i*LOVECLIM. Moreover, our results show that the timing of the first response to the iceberg forcing coincides between all the experiments in the various regions and happens within 300 years. Furthermore, the experiments display two main patterns in the  $\delta^{18}O_{calcite}$  signal. On the one hand, the central and northeast North Atlantic regions display almost no response in  $\delta^{18}O_{calcite}$  to the applied iceberg forcing since the changes in sea sur-
- face temperature and  $\delta^{18}O_{seawater}$  compensate each other or, if the forcing is applied long enough, a delayed response is seen. On the other hand, we show that in Baffin Bay, the Nordic Seas and the subtropical North Atlantic the change in  $\delta^{18}O_{seawater}$  exceeds the sea surface temperature signal and there the  $\delta^{18}O_{calcite}$  pattern closely follows the  $\delta^{18}O_{seawater}$  signal and displays a continuous decrease over the length of the Heinrich event with the minimum value at the end of the iceberg
- release. The comparison of the model experiments with four marine sediment cores indicates that the experiment with an iceberg forcing of 0.2 Sv for 300 years yields the most reasonable results.

# 1 Introduction

The current climatic warming is causing accelerated mass loss from the Greenland and Antarctic ice sheets (e.g. Rignot et al., 2011), producing enhanced runoff and iceberg calving, and increased fresh-

25 water fluxes into the oceans (e.g. Vizcaíno et al., 2008). In the Northern Hemisphere, higher freshwater fluxes might weaken the Atlantic Meridional Overturning Circulation (AMOC), thus altering the global climate (Manabe and Stouffer, 1995; Ganopolski and Rahmstorf, 2001). The interactions between freshwater fluxes and the AMOC are however complex, since the AMOC's response depends among other things on the amount and the exact location of the added freshwater fluxes (Roche et al.,

2004; Swingedouw et al., 2009; Smith and Gregory, 2009). This complexity has been addressed in numerous studies using either numerical models, or proxy data or a combination of the two methods (e.g. Stommel, 1961; Manabe and Stouffer, 1995; Rahmstorf, 2002; Stouffer et al., 2006; Roche et al., 2010).

Proxy data have provided convincing evidence of rapid climate changes in the past that were related to a slowdown or even shut-down of the AMOC (Broecker et al., 1985). Proxy data, such as collected in ocean cores, provide information about the ocean's state during past times. For example, the isotopic analysis of benthic (bottom dwelling) and planktonic (surface dwelling) foraminifera gives information on the temperature and isotopic composition of seawater during the formation of their shells (Urey, 1947; Shackleton, 1974). Furthermore, knowledge about the ocean's ventilation

- can be gathered from benthic foraminifera δ<sup>13</sup>C (Lynch-Stieglitz et al., 1995) and the strength of the ocean circulation can be estimated from the sedimentary protactinium-thorium ratio (Pa/Th, Yu et al., 1996; Balcerak, 2011). Unfortunately, these proxies do not only react to changes in ocean circulation, but also to changes in biological productivity (in the case of benthic foraminifera δ<sup>13</sup>C, Lynch-Stieglitz et al., 1995) or to changes in particle fluxes (in the case of Pa/Th, Geibert and Usbeck,
- 2004) complicating the interpretation of the recorded signal. Nevertheless, they provide valuable insight into past climate conditions and changes, such as a strong reduction in oceanic circulation during Heinrich stadial 1 (HS1), that is the time period between ~ 17.5 calendar ky BP (ka) and the transition towards the Bolling-Allerod warm interval at 14.7 ka (McManus et al., 2004; Gherardi et al., 2005, 2009).
- Heinrich events are specific episodes during which large armadas of icebergs were released from the Northern Hemisphere ice sheets, leading to widespread sedimentation of ice-rafted debris (IRD) across the high-to-mid latitude North Atlantic Ocean (e.g. Andrews et al., 2000; Hemming, 2004). Heinrich events are thus characterized by a steep increase of ice rafted debris in North Atlantic sediment cores and six of those events were identified during the last glacial cycle (Heinrich, 1988).
- IRD is defined as sediment coming from ice sheets and transported by sea ice and icebergs. The quantity of IRD of a certain grain size fraction (e.g.  $

analysis of Northern-Hemisphere proxies indicates that Heinrich events coincide with cold climate conditions and a weakened AMOC (Hemming, 2004).

- IRD records provide a clear signal of the occurrence of Heinrich events. Yet, the interpretation is not as straightforward for the carbonate oxygen isotopic composition recorded in planktonic foraminifera because it depends on both the prevailing temperature and isotopic composition of seawater. The cooling of the surface ocean on the one hand, the freshening and the associated depletion of the seawater on the other hand, might cancel each other out or delay the signal in the  $\delta^{18}O_{calcite}$ .
- Using an isotope-enabled climate model, Roche and Paillard (2005) investigated the contribution of these two effects to the  $\delta^{18}O_{calcite}$  signal. They showed that as a result of the competing effects, the modeled  $\delta^{18}O_{calcite}$  at the location of the ocean core MD95-2042 displays a weaker signal during Heinrich event 4 (~40 ka) than expected, which fits reasonably to the corresponding proxy data (Shackleton et al., 2000; Roche and Paillard, 2005).
- Overall, it has proven difficult to constrain the duration and amount of released freshwater during Heinrich events. However, climate models are helpful tools for this purpose. Estimates derived from proxy data and climate models range from 250 to more than 1250 years with yearly iceberg melt fluxes of 0.04 Sv to 0.4 Sv (1 Sv =  $10^6 \text{ m}^3 \text{ s}^{-1}$ , Hemming, 2004; Roche et al., 2004; Levine and Bigg, 2008; Green et al., 2011; Jongma et al., 2013; Roberts et al., 2014). In particular, Roche et al.
- (2004) showed that the most likely scenario for Heinrich event 4 was a duration of  $250 \pm 150$  years with a meltwater flux of  $0.29 \pm 0.05$  Sv using a climate model with an isotope-enabled ocean model. Recently, Roche et al. (2014) found that the best fit between modeled and observed  $\delta^{18}O_{calcite}$  of Heinrich event 1 (~17.5 ka) is achieved when the AMOC is strongly weakened, but not completely shut down. These authors used an isotope-enabled climate model to perform hosing experiments,
- thus adding the freshwater related to iceberg discharge directly to the ocean, and compared the simuated  $\delta^{18}O_{calcite}$  of the model with proxy data at various ocean depths. Moreover, they showed that the best agreement between model and data is found when freshwater is added in the Labrador Sea, thus mimicking icebergs calved from the Laurentide ice sheet rather than dumping the water in the Ruddiman belt (40° -55° N, Ruddiman, 1977) as has been done before (e.g. Schmittner et al.,
- 2002; Timmermann et al., 2005; Hewitt et al., 2006). Another approach to constrain the freshwater flux released during Heinrich events was taken by Roberts et al. (2014), who simulated the sediments discharge of icebergs using an active iceberg model to mimic the IRD found in ocean cores. Their set-up indicates a much weaker freshwater flux of 0.04 Sv over 500 years than expressed by previous studies, but the authors notice that the total volume released is similar to the one obtained by Roche et al. (2004).

Other modelling studies have concentrated on explicitly simulating the impact of icebergs on climate during Heinrich events (Levine and Bigg, 2008; Green et al., 2011; Jongma et al., 2013). These studies found that simulating Heinrich events using a climate model with an implemented interactive iceberg model results in a different AMOC response compared to hosing experiments.

Hosing experiments ignore the take up of latent heat needed to melt the icebergs and underestimate the spread of the meltwater anomaly. Both effects cause a different ocean state than directly applying freshwater fluxes (e.g. Jongma et al., 2009; Bügelmayer et al., 2015a).

To summarize, the modeling approaches taken so far either concentrated on determining the duration and freshwater fluxes by comparing the modeled isotope or IRD values with proxy data (Roche

- et al., 2004, 2014; Roberts et al., 2014) or on the explicit computation of the icebergs' effect on the ocean's state (Levine and Bigg, 2008; Green et al., 2011; Jongma et al., 2013). In the present study we combine the two approaches of isotopic modeling and interactively computed icebergs by using a global isotope-enabled climate iceberg model. We concentrate on Heinrich Event 1 (~17.5-14.7 ka), thereby extending the work of Roche et al. (2014), who showed that the freshwater flux that
- yields model results in best agreement with available proxy data evidence is 0.2 Sv. Moreover, it allows us to use their tested and described background state of the Last Glacial Maximum (LGM, 21 ka, Roche et al., 2014).

Using the *i*LOVECLIM climate model we aim to investigate the following research questions:

(1) what is the impact of the duration of the iceberg discharge on the climate's response? (2) To what
 extent does the simulated signal in δ<sup>18</sup>O<sub>calcite</sub> during Heinrich event 1 depend on its location? (3)
 How do the changes in ocean temperatures and δ<sup>18</sup>O<sub>seawater</sub> caused by the iceberg discharge and related changes in ocean circulation impact the δ<sup>18</sup>O<sub>calcite</sub> recorded in proxies?

### 2 Methods

### 2.1 *i*LOVECLIM climate model

- The climate model *i*LOVECLIM version 1.0 is an earth system model of intermediate complexity and a code fork of the LOVECLIM 1.2 model (Goosse et al., 2010). *i*LOVECLIM has been further developed to include the explicit computation of water isotopes in the atmosphere, ocean, continental surface (Roche, 2013) and icebergs. Following the description of Bügelmayer et al. (2015a) we state the main characteristics of the ocean, atmosphere and vegetation models.
- The included atmospheric model ECBilt (Opsteegh et al., 1998) is a quasi-geostrophic, spectral model calculated with a time step of 4 hours on a horizontal T21 truncation (5.6° in latitude/longitude) and three vertical pressure levels (800, 500, 200 hPa). The cloud cover is prescribed according to climatology (ISCCP D2 dataset, Rossow et al., 1996) and precipitation is only computed in the lower most (tropospheric) layer and is obtained using the available humidity at this level. To compute the
- soil moisture, a simple bucket model is implemented that takes into account evaporation, precipitation and snow melt. If the water content exceeds a defined threshold, the excess water is automatically transported as runoff to the corresponding ocean grid point. The sea-ice and ocean component CLIO consists of a dynamic - thermodynamic sea-ice model (Fichefet and Maqueda, 1997, 1999) coupled to a 3D ocean general circulation model (Deleersnijder and Campin, 1995; Deleersnijder

- et al., 1997; Campin and Goosse, 1999). The discretization is done on an approximately 3° x3° in longitude and latitude and presents 20 unevenly spaced vertical levels in the ocean. The formulation of the surface albedo of the sea ice takes into account its state (frozen or melting) and the thickness of the snow and ice covers (Goosse et al., 2010). The ocean model has a free surface allowing the use of real freshwater fluxes and a realistic bathymetry. The vegetation model used is VECODE
- (Brovkin et al., 1997) that accounts for two plant functional types (trees and grass) and bare soil as a dummy type. It has the same resolution as the atmospheric model, but allows fractional use of one grid cell to consider small spatial changes in vegetation. It depends on the temperature and precipitation provided by ECBilt and accounts for long-term (decadal to centennial) changes of the climate.
- Icebergs are computed using the optional dynamic thermodynamic iceberg module included in *i*LOVECLIM (Jongma et al., 2009; Wiersma and Jongma, 2010), which is based on the icebergdrift model of Smith and co-workers (Smith and Banke, 1983; Smith, 1993; Løset, 1993) and on the developments done by Bigg et al. (1996, 1997) and Gladstone et al. (2001). According to the provided ice mass icebergs of 10 size classes are generated at the pre-defined calving locations
- following the size distribution presented by Bigg et al. (1996) and based on present day observations (Dowdeswell et al., 1992). We do not expect to introduce a strong bias due to the use of the presentday distribution under LGM conditions, because the chosen size classes only have a marginal impact on the resulting ocean state (Bügelmayer et al., 2015b). Icebergs are moved by the Coriolis force, the air-, water-, and sea-ice drag, the horizontal pressure gradient force and the wave radiation force.
- The icebergs melt over time due to basal melt, lateral melt and wave erosion and may roll over as their length to height ratio changes. The heat needed to melt the icebergs is taken from the ocean layers corresponding to the icebergs' depth, and their meltwater fluxes are put into the ocean surface layer of the current grid cell. The refreezing of melted water and the break-up of icebergs are not included in the iceberg module.
- In ECBilt water isotopes are treated in the same way as moisture to ensure consistency between freshwater fluxes and isotopic fluxes (Roche, 2013). In CLIO water isotopes are handled as passive tracers. In the current set-up, isotopes have also been added to the iceberg module with a fixed value of -30‰chosen to mimic the depletion of the ice-sheet source and the exact value chosen is not important for the current study. The freshwater flux and the water isotopes are accordingly added to the oceans' surface layer when the icebergs melt.
- ob the occans surface hayer when the record

# 2.2 Experimental set-up

Using the *i*LOVECLIM model, Roche et al. (2014) found the best agreement between simulated and observed  $\delta^{18}O_{calcite}$  of Heinrich event 1 when applying a strong freshwater forcing of about 0.2 Sv over 300 years. This value depends of course on the model and the duration of the applied forcing.

Moreover, the modeled  $\delta^{18}O_{calcite}$  fits better to proxy data when the freshwater forcing was applied in the Labrador Sea than when added in the Ruddiman belt (Roche et al., 2014).

We therefore use a similar set-up as Roche et al. (2014) and generate icebergs of a total volume of  $1.7*10^{10}$  m<sup>3</sup> year<sup>-1</sup>, which corresponds to a 0.2 Sv freshwater flux. The icebergs are released in the Labrador Sea (Figure 1). We performed three experiments, all started from the same initial conditions as described in (Roche et al., 2014), but conducted over different lengths to test the impact

- conditions as described in (Roche et al., 2014), but conducted over different lengths to test the impact of the duration of the iceberg release on the oceanic conditions. First, we performed the so-called ICE-300 experiment where we released an iceberg flux of 0.2 Sv for 300 years; second, the ICE-600 where the iceberg flux of 0.2 Sv was applied for 600 years; and third, the ICE-900 with an iceberg flux of 0.2 Sv for 900 years. All the experiments were started from the same initial conditions and
- the first 100 years were modeled without icebergs and are further used as a control state. After the iceberg forcing stopped, the experiments were conducted for another 600 years, thus the ICE-300 was integrated over a total of 1,000 years, ICE-600 over 1,300 model years and ICE-900 over 1,600 model years.

# 3 Results

## 180 3.1 Climate response to iceberg forcing

The AMOC strength is greatly weakened in all experiments, independently of the duration of the applied iceberg flux (Figure 2). This reduction is related to the weakening of the deep convection by the melting icebergs, especially in the Labrador and Greenland - Iceland - Norwegian (GIN) Seas. After 300 years of iceberg melt flux, the AMOC is strongly reduced, but not shut down. This is also seen in the fact that it starts to recover in ICE-300 as soon as the iceberg discharge stops (Figure 2, green line). In ICE-600 and ICE-900 however, the persistent supply of icebergs causes the AMOC to shut down, therefore it needs another 700 years until it starts to recover in ICE-600 and 2,200 years in ICE-900 (not shown). This demonstrates that in *i*LOVECLIM the ability of the AMOC to recover from a Heinrich event-like iceberg discharge is non-linear and strongly depends on the duration of

190 the applied freshwater fluxes.

The duration of the iceberg release impacts the maximum amplitude of the oceanic changes, but the timing of the first response in sea surface temperature (SST),  $\delta^{18}O_{seawater}$  and  $\delta^{18}O_{calcite}$  is identical in all the experiments and happens within the first 300 years of iceberg discharge. Therefore, only the results of the ICE-300 set-up are shown in Figure 3. As expected, the change in SST,

$\delta^{18}O_{seawater}$  and  $\delta^{18}O_{calcite}$  caused by the icebergs is immediate at the calving locations and in the Ruddiman belt (Figure 3) due to the strong iceberg melt flux (IMF). In these areas, SST and  $\delta^{18}O_{seawater}$  decrease significantly within 0-2 years of the icebergs release, which is also seen in  $\delta^{18}O_{calcite}$  (Figure 3). Other regions, such as the Greenland Sea or the Arctic Ocean respond up to 70 years after the start of the iceberg discharge due to the slow advection of the icebergs meltwater

into these regions. Overall the timing of the  $\delta^{18}O_{calcite}$  signal is closer to timing of the SST than of  $\delta^{18}O_{seawater}$  signal, which indicates the stronger impact of SST on the  $\delta^{18}O_{calcite}$  signal (Figure 3). However, in Baffin Bay the  $\delta^{18}O_{seawater}$  is significantly altered by the depleted IMF within 10 years, as also seen in  $\delta^{18}O_{calcite}$ , yet the SST does not display any change or only after up to 70 years. This is due to the prevailing cold conditions and to the presence of sea ice in Baffin Bay 205 (Figure 4b, e), which prevent the SST to strongly react to the IMF.

All the experiments display the maximum IMF in the Labrador Sea due to the located calving sites (Figure 1) and in the Ruddiman belt (Figure 5, Figure 6, Figure 7,a). In the latter area, we also find the minimum sea surface temperatures (Figure 5, Figure 6, Figure 7, b) at the end of the iceberg release caused by the combination of the icebergs cooling effect and the reduced ocean

- circulation (Figure 5, Figure 6, Figure 7, c). In *i*LOVECLIM, the major convection sites are situated in the Labrador Sea and Nordic Seas, southeast of the sea ice margin (Figure 4a, c; CLD=convection layer depth) during LGM conditions, as in other studies (e.g. Dokken and Jansen, 1999; Seidov and Maslin, 1999; Hewitt et al., 2001). The IMF strongly impacts the deep convection in the Labrador Sea and also in the Nordic Seas, especially in ICE-600 and ICE-900. The location and magnitude of
- the strongest change in SST is almost identical in all three experiments, but in ICE-600 and ICE-900 the longer duration of the freshwater forcing causes a wider spread cooling than seen in ICE-300 (Figure 5, Figure 6, Figure 7, b). Also the locations of the minimum  $\delta^{18}O_{seawater}$  at the end of the freshwater flux coincide within the experiments (Figure 5, Figure 6, Figure 7,d) as it decreases greatly at and close to the calving locations in the Labrador Sea. Furthermore, the advection of
- iceberg melt flux causes minimum  $\delta^{18}O_{seawater}$  values at the sea surface northeast of Greenland, especially in ICE-600 and ICE-900 (Figure 5, Figure 6, Figure 7,d). The response in sea surface salinity greatly varies between the three experiments south of 45° N and in the Nordic Seas (Figure 5, Figure 6, Figure 7,e). In these two regions ICE-600 and ICE-900 show much stronger reductions in SSS than ICE-300 at the end of the iceberg discharge because there the decrease is mainly caused by
- the advection of the fresh surface waters rather than by the amount of icebergs reaching these areas, which is comparable in all three experiments.

Also, the  $\delta^{18}O_{seawater}$  and the  $\delta^{18}O_{calcite}$  signal in the North Atlantic and the Nordic Seas (Figure 5, Figure 6, Figure 7, d, f) vary between the three set-ups. ICE-300 displays positive  $\delta^{18}O_{calcite}$  values south of 45° North and in the Nordic Seas, but in ICE-600 these are limited to a small region in the

- North Atlantic and the ICE-900 run displays purely negative values (Figure 5, Figure 6, Figure 7,f). The lower  $\delta^{18}O_{seawater}$  and  $\delta^{18}O_{calcite}$  signals in ICE-600 and ICE-900 result from the combined effect of the isotopically depleted iceberg melt flux directly released by the icebergs and the slow advection of fresh surface waters into the Nordic Seas and further south (Figure 5, Figure 6, Figure 7,e).
- To summarize, in our model, the length of the iceberg discharge (300, 600 or 900 years) strongly impacts the AMOC's ability to recover after the Heinrich event. However, the spatial patterns of

the deep convection and the sea surface temperature at the end of the Heinrich event-like iceberg release coincide well within the three experiments. Yet, the response in sea surface salinity, δ<sup>18</sup>O<sub>seawater</sub> and δ<sup>18</sup>O<sub>calcite</sub> depends on the duration of the iceberg release, especially in the North
 Atlantic. Moreover, we find an immediate response in δ<sup>18</sup>O<sub>calcite</sub> to the iceberg release at the calv-

ing sites and in the North Atlantic, but it takes more than 100 years to cause a significant change in regions further away from the calving sites.

As a next step, we analyze in detail the evolution in  $\delta^{18}O_{calcite}$  over the course of a Heinrich event in the regions of largest negative SST and  $\delta^{18}O_{seawater}$  anomalies, since  $\delta^{18}O_{calcite}$  is primarily

impacted by these two variables (Shackleton, 1974). As described above, the minimum SSTs are found in the North Atlantic due to the combination of IMF and decreased convection layer depth, the areas of minimum  $\delta^{18}O_{seawater}$  are on the one hand in Baffin Bay, at the calving sites, and on the other hand in the Nordic Seas, due to the advection of depleted surface waters.

# 3.2 Minimum SST - its effect on the $\delta^{18}O_{calcite}$ evolution

- Concentrating on the Central North Atlantic region as defined in Figure 8, which displays the minimum SSTs in all three experiments, independently of the duration of the iceberg release (300 vs 600 vs 900 years), we see that the time evolution of the IMF clearly displays a high, but slightly varying flux (1000 to 1200  $\text{m}^3 \text{ s}^{-1}$ , Figure 9a). This region experiences high IMF values because most icebergs are transported there and the ocean conditions cause them to melt. The values are sim-
- ilar in all three experiments, but the response in ocean temperature and  $\delta^{18}O_{seawater}$  clearly differ (Figure 9a). Concerning the SSTs, a cooling of about 6-7° C is seen in all three experiments, but in ICE-300 the SST increases towards its initial value as soon as the freshwater forcing stops (Figure 9a), which is not the case in ICE-600 or ICE-900 due to the severely disturbed AMOC. The signal in  $\delta^{18}O_{seawater}$  reflects the input of the isotopically depleted iceberg melt water (-30 %c) that causes
- a decrease in  $\delta^{18}O_{seawater}$  of 1.6%(ICE-300) to 2.5%(ICE-900). The  $\delta^{18}O_{calcite}$  signal reflects both the temperature and  $\delta^{18}O_{seawater}$  signal. Therefore, in ICE-300 and during the first 300 years of freshwater input in ICE-600 and ICE-900 the Heinrich event is characterized by an increase in  $\delta^{18}O_{calcite}$  at the beginning of the forcing. This is caused by the decrease in temperature and displays the counteracting effects of the temperature and  $\delta^{18}O_{seawater}$  (Figure 9a).  $\delta^{18}O_{calcite}$  decreases to-
- wards its initial value when SST and  $\delta^{18}O_{seawater}$  recover. Also the ICE-600 and ICE-900 display the increase in  $\delta^{18}O_{calcite}$  at the beginning of the iceberg release, yet, the continuous supply of low  $\delta^{18}O_{seawater}$  results in a stronger decrease of  $\delta^{18}O_{seawater}$  than in ICE-300, which causes the  $\delta^{18}O_{calcite}$  to decrease again after about 300 years of freshwater pulse (Figure 9a). Both the sea surface salinity and  $\delta^{18}O_{seawater}$  reach their minimum towards the end of the iceberg release and
- increase slightly afterwards without returning to their initial value due to the presence of depleted melt water and reduced deep overturning.

We therefore conclude that at the location of minimum SST, the signal in  $\delta^{18}O_{calcite}$  can on the one hand display an increase and a stable phase over the length of the Heinrich event due to the competing effects of the decreasing temperature and  $\delta^{18}O_{seawater}$ . On the other hand, if the freshwater pulse is applied long enough, the initial increase is followed by a decrease as soon as the amplitude of the decreased  $\delta^{18}O_{seawater}$  exceeds the drop in temperature.

# 3.3 Minimum $\delta^{18}O_{seawater}$ - its effect on the $\delta^{18}O_{calcite}$ evolution

- As expected at the calving sites, the IMF is constant and relatively high in Baffin Bay over the whole duration of the Heinrich event (Figure 9b). The take-up of heat needed by the icebergs to melt causes an immediate drop of about  $1.5^{\circ}$  C in temperature at the ocean's surface until it reaches its freezing point of about  $-2^{\circ}$  C. Note that the initial rise in SST over the first 100 years is due to internal variability. The SSS responds with a continuous decrease over the length of the freshwater forcing with the minimum occurring at the end of the iceberg release (Figure 9b). The  $\delta^{18}O_{seawater}$  also
- steeply decreases as soon as the icebergs start to melt and then continues to decrease over the length of the forcing. In ICE-300, at the end of the iceberg release, the  $\delta^{18}O_{seawater}$  increases towards its initial value and reaches it after about 300 years. In ICE-600 and ICE-900, however, it increases by about 2‰but does not reach its initial value because the long duration of the iceberg release caused a fresher ocean state compared to before the iceberg discharge. The  $\delta^{18}O_{seawater}$  pattern is mimicked
- by the  $\delta^{18}O_{calcite}$ , displaying the fact that the change in temperature is of smaller magnitude than that in  $\delta^{18}O_{seawater}$  (Figure 9b).

Due to the presence of sea ice, the IMF reaching the Nordic Seas is not larger than 10 to  $25m^3 s^{-1}$  (Figure 9c), since the few icebergs that reach that far North are pushed southward again by the ice and the wind drag (not shown). Yet, as in Baffin Bay, the temperature quickly drops by about  $3^{\circ}$  C

- and the SSS starts to decrease because the advection of the cold and fresh surface water enhances the response in SST and SSS. The  $\delta^{18}O_{seawater}$  begins to drop as soon as the icebergs are released, but steepens its curve later on due to the supply of surface waters (Figure 9c). As seen in Baffin Bay, the Nordic Seas  $\delta^{18}O_{calcite}$  pattern closely resembles the  $\delta^{18}O_{seawater}$  signal (Figure 9c), except within the first 20 years of the iceberg discharge, where the strong drop in SST postpones its decrease.
- We find that overall the time series of the  $\delta^{18}O_{calcite}$  signal is very similar between the two regions of maximum  $\delta^{18}O_{seawater}$  decrease, mimicking the  $\delta^{18}O_{seawater}$  pattern rather than the SST. However, the  $\delta^{18}O_{seawater}$

response at the start of the iceberg discharge depends on the mechanisms at work, where in Baffin Bay the drop is immediate due to the calving locations, while in the Nordic Seas it takes about 20 years until the cold and fresh surface waters are advected into this region.

We conclude that in regions of weak surface cooling, but strong  $\delta^{18}O_{seawater}$ 

changes, the  $\delta^{18}O_{calcite}$  mimics the  $\delta^{18}O_{seawater}$  pattern. The mechanisms causing the drop in

 $\delta^{18}O_{seawater}$  (freshwater as in Baffin Bay or advection of depleted surface waters as in the Nordic Seas) determine the timing of its response at the beginning of the Heinrich event.

# 310 3.4 How does the $\delta^{18}O_{calcite}$ evolution look in other regions?

In regions where neither the temperature nor the  $\delta^{18}O_{seawater}$  signal dominates, we find that the  $\delta^{18}O_{calcite}$  evolution can represent either a stable phase, displaying counteracting effects of SST and  $\delta^{18}O_{seawater}$  or mimic the  $\delta^{18}O_{seawater}$  (Figure 9d,e). In the Northeast North Atlantic the temperature and  $\delta^{18}O_{seawater}$  decrease almost balance each other in the ICE-300 experiment (Figure 9d),

but in ICE-600 and ICE-900 the continuous decrease in  $\delta^{18}O_{seawater}$  exceeds the temperature signal after about 400 years, causing a decrease in  $\delta^{18}O_{calcite}$ , similar to the pattern seen in the central North Atlantic (Figure 9a).

Almost no IMF reaches the subtropical North Atlantic, only occasional pulses of  $\sim 0.5 \text{ m}^3 \text{ s}^{-1}$ . Nevertheless, the weakened AMOC causes the SST to decrease about 10-30 years after the beginning

of the Heinrich event as less warm waters from the Southern Hemisphere are transported into this region. However, it takes almost 300 years of iceberg discharge to transport the fresh melt water that far south to result in a decreasing SSS (Figure 9e). In ICE-300 the  $\delta^{18}O_{seawater}$  shows a slow decrease of about 1%cover the length of the forcing, causing also a weak response in  $\delta^{18}O_{calcite}$ . The steady supply of icebergs in ICE-600 and ICE-900 cause a much stronger decrease in SSS,

$\delta^{18}O_{seawater}$  and  $\delta^{18}O_{calcite}$  than in ICE-300, which indicates the importance of the length of the Heinrich event (Figure 9e).

Overall, we find two main patterns in  $\delta^{18}O_{calcite}$  in our model simulations. First, we find no or a delayed decrease in  $\delta^{18}O_{calcite}$  in the central and northeast North Atlantic regions, where the changes in SST and  $\delta^{18}O_{seawater}$  compensate each other. Second, the  $\delta^{18}O_{calcite}$  pattern closely

follows that of  $\delta^{18}O_{seawater}$  in Baffin Bay, the Nordic Seas and the subtropical North Atlantic, where the change in  $\delta^{18}O_{seawater}$  exceeds the SST signal, and displays a continuous decrease over the length of the Heinrich event with the minimum value at the end of the iceberg release.

#### 4 Discussion

# 4.1 Can we use the modeled $\delta^{18}O_{calcite}$ evolution to better understand the

# $\delta^{18} \mathrm{O}_{calcite}$ recorded in marine sediment cores?

We compared our model results to planktonic  $\delta^{18}O_{calcite}$  data from four marine sediment cores distributed over a wide spatial area of the North Atlantic and Greenland - Iceland - Norwegian (GIN) Seas (Figure 8). These cores were selected because of the high temporal resolution of their planktonic  $\delta^{18}O$  records (time step of *sim* 200 years on average) and very good dating control. Two cores

(NA87-22, CH69-K09) are situated within the Ruddiman Belt, one core (ENAM93-21) is located in the Norwegian Sea and the fourth core (KNR31 GPC-5) is relatively far south of Greenland

 $(33^{\circ} \text{ N}, \text{Figure 8})$ . For cores NA87-22 and CH69-K09  $\delta^{18} \text{O}_{calcite}$  was measured on two species: *G. bulloides* and *N. pachyderma* s. *N. pachyderma* s. is expected to prefer a deeper living habitat than *G. bulloides* and depending on the stratification of the water column the two species display a uniform

- (well mixed) or non-uniform pattern (stratified water column; Kohfeld et al. (1996); Simstich et al. (2003)). SST reconstructions derived from planktonic foraminifer counts and IRD data of these two c