# Peer review of "Disentangling the effect of ocean temperatures and isotopic content on the oxygen - isotope signals in the North Atlantic Ocean during Heinrich Event 1 using a global climate model"

_Climate of the Past, 2016_

## Referee Comment (RC1) · Anonymous Referee #1 · 6 Apr 2016

This study analyses three simulations of Heinrich Event 1 with an isotope-enabled model of intermediate complexity. Time series of simulated d18O seawater and calculated d18O calcite (based on simulated d18O seawater and simulated seawater temperatures) are then analysed and loosely compared to four sediment cores. The three different simulations differ in the length of the iceberg calving episode and have different impacts on the Atlantic Overturning Motion. The authors conclude that the duration of the simulated Heinrich event causes a strong and non-linear response in the AMOC.

I have several major problems with this study.

First, a very similar study appeared last year (Bagniewski et al., 2015, "Quantification of factors impacting seawater and calcite d18O during Heinrich Stadials 1 and 4", Paleoceanography, 30(7), 895-911). Bagniewski et al. conducted several simulations of Heinrich Events with an isotope-enabled model of intermediate complexity. They disentangled the effect of ocean temperature and isotopic content on d18O calcite in the North Atlantic and worldwide. They took this study one step further and also disentangled the pure d18O seawater meltwater signal and the changes in d18O seawater due to changes in circulation and climate. They then compared their simulated values with over 20 sediment records at the surface and at depth (including time series in their supplementary material). Bagniewski et al.'s research questions and conclusions are similar to what is presented here.

There is however one exciting new aspect in this study: the simulation of more realistic meltwater pattern and the fact that seawater temperature changes due to latent heat loss of melting icebergs are taken into account. To make this study more original, one solution might be to rewrite the paper and focus on the impact of a more realistic parameterization of iceberg drift and melting on d18O seawater and d18O calcite (i.e. compare pure hosing experiments with the same amount of total freshwater with the simulations presented here). Given that most modeling studies of Heinrich events are based on simplified meltwater hosing scenarios, such a study would be of great interest to the modeling community. An in-depth discussion on how well such a coarse resolution model can be expected to capture iceberg drift should also be included.

Second, the authors cannot conclude that it is the length of the Heinrich Event that causes the non-linear response of the AMOC. To make this conclusion, one additional simulation should be integrated which introduces the same overall volume of icebergs released in ICE-900 over only 300 years. I would not be surprised if the impact on the AMOC in such a simulation is similar to ICE-900, but I might be wrong. There are many publications discussing the hysteresis behavior of the AMOC for a whole zoo

of models (including model intercomparison studies), and I would strongly encourage the authors to cite the original literature when discussing this aspect. Most of these publications base their analysis on the total volume of freshwater added. If LOVECLIM shows a significantly different behavior based on the duration only (and not on total volume) then this might be an interesting and publishable result. Otherwise, there is nothing new about this result.

Third, the comparison with only 4 sediment cores is deceptive. There are more cores in the North Atlantic that recorded d18O calcite changes over this period with high enough sedimentation rates. These cores should be taken into account in this analysis.

Finally, time series of the model results at the location of these cores should be shown with the sediment data in the same figure (for the better or worse) and discrepancies should be discussed. Nobody would expect a perfect fit, but this one-on-one comparison is still important to validate the model simulations.

Other comments:

Proxy data reconstructions and a few model simulations suggest that the main source of dense waters in the North Atlantic during the last glacial was the Nordic Seas. Labrador Sea Water seems to have played a small (or maybe even absent) role. LOVE-CLIM simulates an important convection site in the Labrador Sea for the control run. During the transient simulations, this convection site shuts off and influences d18O seawater, temperatures and, of course, the strength of the AMOC discussed in this paper. The fact that this convection site might or might not be realistic and how this impacts the results should be discussed in the paper.

Along the same lines, wouldn't one expect Baffin Bay to be completely ice covered and at freezing point just before and during Heinrich Event 1? How realistic are the simulated warm conditions in this region (probably related to the near-by convection site) at this point of time? See for example Gibb et al. 2015, "Diachronous evolution of sea surface conditions in the Labrador Sea and Baffin Bay since the last deglaciation",

The Holocene, 25(12), 1882–1897.

What is the equivalent sea level rise for each of the 3 simulations and how does this compare to data/reconstructions?

Which equation is used to calculate d18O calcite in the model?

Line 36: "ocean cores" should probably read "sediment cores"

Line 55: IRD transported by sea ice?

Line 56: Should read ">63um"?

Line 158-159: How can the value of iceberg d18O not be important for a study that analyses changes in d18O during a Heinrich event?

Line 187: it is not obvious (Fig 2) that ICE-600 recovers 700 years later

---

## Referee Comment (RC2) · Anonymous Referee #2 · 17 Apr 2016

Comments on manuscript cp-2016-31 "Disentangling the effect of ocean temperatures and isotopic content on the oxygen - isotope signals in the North Atlantic Ocean during Heinrich Event 1 using a global climate model" by M. Bügelmayer-Blaschek, D. M. Roche, H. Renssen, and C. Waelbroeck

**General comments**

With an Earth system model of intermediate complexity including an iceberg module the authors investigate the distributions of $\delta^{18}O$ of water and of calcite in the North Atlantic ocean during Heinrich event 1. They analyze the temporal evolution of $\delta^{18}O_{calcite}$ and put forward two different geographical patterns: areas where the $\delta^{18}O_{calcite}$ hardly changes (or with large delay) during H1 in contrast to other areas where the $\delta^{18}O_{calcite}$ closely mimics the evolution of that of $\delta^{18}O_{seawater}$.

This is a very interesting research subject which is helpful in the context of improving our understanding of past climates. The method and tools are pertinent. However the analysis of the results is somewhat too qualitative. The draft seems to have been hastily written with several repetitions and inconsistencies. If re-worked thoroughly this could become a very pertinent paper.

Subject to the revisions of the specific comments below I would recommend publication in CP.

**Specific comments**

1. One important aspect which is not addressed in this study is whether the inclusion of an iceberg module in the model does help improving the modeling of $\delta^{18}O$ during Heinrich events or not? In short: is it worth including icebergs in climate models? Does it bring significant improvement of modeling studies? A comparison with the results of already available water hosing experiments performed with the same model would be welcome and significantly add to the value of the present work.

2. Some of the conclusions are clearly overstated: "The comparison of the model experiments with four marine sediment cores indicates that the experiment with an iceberg forcing of 0.2 Sv for 300 years yields the most reasonable results." (lines 20-21), "we find that the set-up of an iceberg forcing of 0.2 Sv over 300 years yields the most reasonable results" (line 403), and "From the comparison of simulated sea surface temperatures and $\delta^{18}O_{calcite}$ with proxy data, we find the best agreement between model output and data is reached when the iceberg discharge is stopped after 300 years." (lines 475-477). The data-model comparison as performed so far does not provide enough evidence for such conclusions.

3. The (short, lines 381-401) data-model comparison carried on page 12 suffers from several shortcomings.

   1. Cores 2 & 3 (NA87-22 & CH69-K09) are compared to the model results in the Baffin Bay (lines 381-386). Yet, if the map in Fig. 8 is correct, cores 2 & 3 are not expected to be representative of the Baffin Bay, but rather of the North-East Atlantic and the central North Atlantic.

   2. ICE-600 et ICE-900 display similar changes in the Nordic Seas as observed in core 3 (ENAM93-21) but this agreement is dismissed on the basis of a lack of agreement for SST which is not represented for that area. Further, this decision somewhat contradicts the (sound) remark that reconstructed SST for high latitudes have high uncertainties

(lines 448-451).

3. We miss the information in panel a) and d) of Fig. 9 to appreciate how the $\delta^{18}O_{calcite}$ from core 4 (KNR316-GPC5) might fit the different experiments.

4. The duration of the perturbation in cores 1, 2, and 3 appear to be much longer than in the model experiments. How do you explain it?

The data-model comparison is minimalist and too qualitative. I would recommend the use of additional cores – many are available – to improve that aspect. Wouldn't it also possible to perform cross-correlation between time series (model versus data; between variables)?

4. "the timing of the first response to the iceberg forcing coincides between all the experiments" (abstract, lines 10-12;); "Yet, our results show that the timing of the first response to the iceberg forcing in SST, $\delta^{18}O_{seawater}$ and $\delta^{18}O_{calcite}$ coincides between all the experiments in the various regions within 300 years." (conclusion, lines 472-474): this should not be presented as an outcome of the study; given the experimental setup this behavior is expected.

5. The discussion on MOC (lines 421-434) should be dropped; it is neither relevant nor needed for the present work.

6. Question "(1) what is the impact of the duration of the iceberg discharge on the climate's response?" (line 109) should be reformulated. The consequences of the icebergs discharge on North Atlantic Ocean properties are examined but their climate impact is nowhere discussed.

7. It is nowhere mentioned how $\delta^{18}O_{calcite}$ is computed.

8. "In these two regions ICE-600 and ICE-900 show much stronger reductions in SSS than ICE-300 at the end of the iceberg discharge because there the decrease is mainly caused by the advection of the fresh surface waters rather than by the amount of icebergs reaching these areas, which is comparable in all three experiments." (lines 223-226). I do not get the argument. Wouldn't it be simply due to the fact that the cumulative freshwater flux is much larger in ICE-600 and in ICE-900? May be reformulate?

9. ..."we find an immediate response in $\delta^{18}O_{calcite}$ to the iceberg release at the calving sites and in the North Atlantic, but it takes more than 100 years to cause a significant change in regions further away from the calving sites" (lines 240-243). This statement appears to be in contradiction with the foregoing discussion and Fig. 3.

**Other comments**

1. line 73: remove "yearly" (brings confusion with Sv units used for fluxes)

2. line 81: "simulated" is missing an "l"

3. lines 87-90: "but the authors notice that the total volume released is similar to the one obtained by Roche et al. (2004)." Based on the numbers given here, the Roche et al. (2004) freshwater volume (0.29 Sv during 250 yr) is nearly 4 times that of Roberts et al. (2014) (0.04 Sv during 500yr). Can you develop the similarity?

4. line 95: replace "the take up" by "the uptake"

5. lines 104-105: the last part of the sentence "who showed that the freshwater flux that yields model results in best agreement with available proxy data evidence is 0.2 Sv" is part of the preceding discussion (lines 70-90) and should be moved there (line 79?)

6. lines 120-121: suggestion "The  atmospheric model ECBilt (Opsteegh et al., 1998) is a quasi-geostrophic, spectral model  running with a time step of 4 hours on a horizontal T21 truncation..."

7. lines 123-124: "...precipitation is only computed in the  lowest (tropospheric) layer..."

8. line 130: suggested changes " CLIO has a resolution of 3°x3°  with 20 unevenly spaced vertical levels in the ocean"

9. lines 135-135: "The vegetation model  VECODE (Brovkin et al., 1997)  accounts for..."

10. lines 137-138: suggested changes " VECODE is forced by temperature  precipitation and $CO_2$ provided by ECBilt and accounts for long-term..."

11. lines 152-153: suggestion "and their meltwater fluxes are put into the ocean surface layer of the current  water column."

12. lines 158 & 399: "‰_", missing space after the per-mil symbol

13. line 164: what does "This value" refer to? Here there should be reference to the work of Kageyama et al. (2010).

14. line 187: the MOC recovery for exp ICE-600 is not seen in Fig. 2

15. lines 188-190: Previous works addressed the issue of MOC recovery in LOVECLIM; e.g., Rahmstrof et al. (2005), Menviel et al. (2008), Kageyama et al. (2010).

16. line 280: suggest " uptake"

17. line 282 suggested "freezing point at  about -2°C"

18. lines 350-352: suggested changes: "Before looking in detail at the four cores to investigate whether or not the simulated patterns can be confirmed by the data, several important remarks need to be made.  First, while the sea level rise due to the released icebergs during HS1 is accounted for in our experiments. , we do not simulate the background sea level rise starting at 19 ka after the onset of the LGM (Lambeck..."

19. lines 395, 399 & 436: ($\sim$ 1Sv)

20. Figures 4 to 7: should specify that $\delta^{18}O_{calcite}$ and $\delta^{18}O_{seawater}$ represent surface values

21. Figures 4 to 7: scales are not legible

22. Figure 9: the left panels are truncated.

23. Figure 9, legend, 3d line: $\delta^{18}O_{sw}$ and $\delta^{18}O_c$ ("c" an "sw" should appear as subscripts).

24. Figure 10: left panels are truncated.

25. Figure 10: unit of IRD in panel a) number of grains/g sediment?

**References**

- Kageyama, M., A. Paul, D. M. Roche, and C. J. Van Meerbeeck (2010), Modelling glacial climatic millennial-scale variability related to changes in the Atlantic meridional overturning circulation: a review. Quaternary Science Reviews, 29, 2931-2956, doi:10.1016/j.quascirev.2010.05.029.
- Menviel, L., A. Timmermann, A. Mouchet, and O. Timm (2008), Meridional reorganizations of marine and terrestrial productivity during Heinrich events, Paleoceanography, 23, PA1203, doi:10.1029/2007PA001445.
- Rahmstorf, S., M. Crucifix, A. Ganopolski, H. Goosse, I. Kamenkovich, R. Knutti, G. Lohmann, B. Marsh, L. Mysak, and Z. Wang, A. Weaver (2005), Thermohaline circulation hysteresis: a model intercomparison. Geophysical Research Letters, 32, L23605, doi:10.1029/2005GL23655.
- Roche, D., D. Paillard, and E. Cortijo (2004), Constraints on the duration and freshwater release of Heinrich event 4 through isotope modelling, Nature, 432, 379–382, doi:10.1038/nature03059.
- Roche, D. M., D. Paillard, T. Caley, and C. Waelbroeck (2014), LGM hosing approach to Heinrich Event 1: results and perspectives from data–model integration using water isotopes, Quaternary Science Reviews, 106, 247-261, doi:10.1016/j.quascirev.2014.07.020.

---

## Referee Comment (RC3) · Anonymous Referee #3 · 10 May 2016

**Disentangling the effect of ocean temperatures and isotopic content on the oxygen - isotope signals in the North Atlantic Ocean during Heinrich Event 1 using a global climate model: a review**

This study exmaines how the simulated delta 18O of calcite varies in a climate model when different time durations of Heinrich Events are simulated. Its results are interesting and present a novel way to think about the signal that is in the available data. It was let down, however, by its poor figures and rather superficial analysis.

I had to work really hard to relate what was going on in the text with what was shown in the figures. Many of the figures (e.g. fig. 10) don't really support the claims that are made in the text: the claims may be true, but I can't satisfy myself that they are from the figures. There are more detailed crticisms later but I urge the authors to think when preparing their figures: how can I make it as easy as possible for a reader to understand my figures. At present I feel that the figures have been prepared with what is easiest for the authors in mind. If your figures are hard to decipher readers won't bother to read the paper, so it really is in your interests to think about this.

There are a number of times in the text (for example the last line of the m/s) where claims are made that the simulated calcite is a "reasonable" fit or other such vague phrases. Simple statistics do exist that can quantitatively define how well series relate to one another. These should be used rather than qualitative claims of "reasonableness". That palaeoclimatology is rife with such vagueness is no excuse for this.

**Specific Comments/questions:**

Motivation:
It would be good to see the motivation for why it was chosen to test the sensitivity of the model to the duration of the simulated Heinrich Events rather than the size. Why vary the duration of the events and not the size? The size (that is flux of ice) and duration are to a certain extent constrained by the total volume of ice that can be released from the Laurentide Ice Sheet, therefore would not a better test to have been to keep the total ice volume constant and thus for the longer duration events have a small flux of ice?

Model description:
Unless I have missed it there is no description of how delta 18O calcite is calculated within the model. This needs to be included along with discussion on how this model quantity relates to the calcite that you might find in forams. For example, is calcite calcuated in a water depth in which *bulloides* lives?

Section 4:
Section 4 could be a very nice section but is currently weak. More effort to link the observations back to the model runs would strengthen this. You are, I think, hindered by the currently horrible figure 10 in doing this.

It would be helpful to make some comments on how well you feel your model could in a perfect world simulate the calcite in forams (see above).

Line 402: "Overall, we see some similarities between the simulated and measured $\delta 18 O$calcite curves and we find that the set-up of an iceberg forcing of 0.2 Sv over 300 years yields the most

reasonable results compared to the proxy data considered"

If I am honest I do not think that I can judge this because comparing the simulated and proxy calcite is near impossible. See my comments on fig. 10.

The phrase "most reasonable" is unacceptably vague.

The regional abbreviations e.g. stNA, neNA, BB etc. are not necessary and detract considerably from the clarity of the manuscript. Please just use the full description.

Other comments

line 475: "From the comparison of simulated sea surface temperatures and δ 18 Ocalcite
with proxy data, we find the best agreement between model output and data is reached when the iceberg discharge is stopped after 300 years."
I'm not sure I know what figure/metric shows this.

**Figure comments.**

Fig 1. Are the calving locations really those weird shapes? Why not use a pseudocolour plot rather than a contouring one to show the location, then one will be able to see what the model grid resolution is.

Figs 5/6/7. These figures should be split up so that all of the panels (a) are on one figure all the (b)s on another etc. In the text the comparisons are always framed in terms of the variable (e.g SST) not the model run (e.g. ICE 600). At present you have to flip between 3 figures for each variable. This is really difficult to do. You shold also make the panels larger: the numbering on the colour scale is invisible.

Fig 8 please put the names of the core on the map.

Fig 9/10. In figure 9 time goes from left to right, in 10 it goes from right to left. So when trying to relate the modelled delta 18O calcite to the proxy delta 18O, not only do you have to flip between figures but you also have to mentally flip them through 180 degrees. Please chose one direction for time and stick to it.

The panels in fig 9 are tiny. On full zoom my nose is about 2cm from the screen when I look at them!! Please make them bigger. The axes are also missing from the leftmost panels.

You must add the simulated calctie 18O curves into fig. 10 so that it is easier to follow the text. Again I have to flip between figure to work out what is happening: this is really hard.

Fig 10. The core that is in panel (a) is the core that is numbered 2 in fig.8. Panel (b) has core one. This is random. Please have panel (a) as core 1, panel (b) core 2 etc. Also as well as the core name put the number in the title.

To correctly interpret fig 10, I have to:

(1) Look at fig8 to work out the number/location of the core,
(2) read the caption to work out what the core name is.
(3) look at fig 9 to find the modelled calcite
(4) look at the title of the plot to find the region
(5) refer back to fig 8 caption because the region names on the fig8 map and in fig9 are different
(6) mentally flip the time axis of the panel in Fig.9
(7) refer back to caption of fig. 10 because I have forgotten which core was which......

This is **7 steps** before I can even look at what the data says.

On the axis in Fig 10 please change the axis label txt to be the same colour as the line to which it refers.

---

## Editor Comment (EC1) · U. Mikolajewicz (Editor) · 17 May 2016

The ms. 'Disentangling the effect of ocean temperatures and isotopic content on the oxygen – isotope signals in the North Atlantic Ocean during Heinrich Event 1 using a global climate model ' by M. Buegelmayer-Blaschek, D. Roche, H. Renssen, and C. Waelbroeck has been seen by three reviewers.

I agree with the reviewers that the material presented in this paper has the potential to

form the base of a very interesting paper definitely worth publishing, but the presentation quality clearly needs to be improved substantially.

Other after my opinion rather important issues raised by the reviewers are:

- The comparison between simulations and proxy records needs to be improved. This is especially true for figures 9 and 10 representing key time series from the simulations and proxy records. It is almost impossible to compare model and data. The use of a different orientation of the time axis in these figures is very disturbing.

- The rather strong focus on the Baffin Bay is not really motivated. In order to simulate a cooling of 1.5K, Baffin Bay must be seasonally ice free. Is this supported by reconstructions?

- There is a lack of a clear motivation why only 4 proxy records have been used (none of them in the most interesting region cNA).

- It is not clear to me, what is the effect of the duration of the prescribed discharge and what is effect of the total amount discharged. In the paper all effects are attributed to the duration. If it is indeed so, some more evidence is needed that the total amount is of less importance.

- The relation used to calculate d18O Calcite from d18O seawater and temperature needs to be given.

- The figures clearly need to be improved.

Please consider all the points raised by the reviewers and respond carefully to each of them.

additional remarks

Many of the plots are difficult to use due to the insufficient plotting.

I personally found the use of different contour levels for the d18O response for sea

water and calcite made it almost impossible to estimate the temperature contribution. Does it make a significant difference whether SST or subsurface temperature is used for the calculation?

In Fig. 4 are the d18O and SSS plots for the North Atlantic almost useless as the whole North Atlantic is only populated by one isoline. The non-linear contour interval focussing on 0 for d18O may be helpful for anomalies but not necessarily for absolute values. Is annual mean CLD really a relevant property? Would not the climatological annual maximum CLD be more relevant allowing to estimate the depth of convection? The gradient of SSS over the glacial North Atlantic is less than 2, there is only the 35 isoline plotted. Is this realistic?

Fig. 9 The panels are much too small. Plotting 3 experiments into one panel with different colours should allow to more clearly see the differences between the experiments. Are all regions necessary?

Only 4 proxy records are selected. Except for neNA they do not match to the designed key regions. This is not particularly helpful for the model/data comparison.

Figures 9 and 10 should be designed in such a way as to make the model/data comparison as easy as possible for the reader.

line 280 How sensitive are the results to the choice of the initial year (e.g. 100 years earlier or later). Has the strong anomalous signal during the first 100 years used as reference period a substantial impact on the anomalies presented in Figs. 5 to 7? Would another 100 years from an unperturbed simulation show a completely ice covered Baffin Bay?

---

## Author Comment (AC1) · 13 Sep 2016

This study analyses three simulations of Heinrich Event 1 with an isotope-enabled model of intermediate complexity. Time series of simulated d18O seawater and calculated d18O calcite (based on simulated d18O seawater and simulated seawater temperatures) are then analysed and loosely compared to four sediment cores. The three different simulations differ in the length of the iceberg calving episode and have different impacts on the Atlantic Overturning Motion. The authors conclude that the duration of the simulated Heinrich event causes a strong and non-linear response in the AMOC.

I have several major problems with this study.

First, a very similar study appeared last year (Bagniewski et al., 2015, "Quantification of factors impacting seawater and calcite d18O during Heinrich Stadials 1 and 4", Paleoceanography, 30(7), 895-911). Bagniewski et al. conducted several simulations of Heinrich Events with an isotope-enabled model of intermediate complexity. They disentangled the effect of ocean temperature and isotopic content on d18O calcite in the North Atlantic and worldwide. They took this study one step further and also disentangled the pure d18O seawater meltwater signal and the changes in d18O seawater due to changes in circulation and climate. They then compared their simulated values with over 20 sediment records at the surface and at depth (including time series in their supplementary material). Bagniewski et al.'s research questions and conclusions are similar to what is presented here.

*We first want to address your concern that our paper is very similar to the Bagniewski et al. (2015). You are right, we unfortunately missed its publication, but we have added the missing citation in our manuscript. We studied the paper by Bagniewski et al. (2015) closely and would argue that it is not very similar, as our study provides important additional insights due to the direct computation of the dynamics of icebergs.*

*It should be noted that our study does not concentrate on the maximum signal of the Heinrich event and how well this is captured in our model, since this was done before by Roche et al (2014). Instead, we were interested in the development of the d18O_calcite's signal at the beginning and during Heinrich event 1, since different factors affect this signal.*

*Please find a detailed list of differences between the two papers in the table below.*

| Bagniewski et al. (2015) | Bügelmayer-Blaschek et al. (2016) |
|---|---|
| **RESEARCH QUESTIONS** | |
| *Analyse the respective impacts of: 1) addition of d18O depleted meltwater in the North Atlantic and its propagation; 2) anomalies in seawater d18O due to changes in ocean circulation, evaporation, precipitation, river discharge, sea ice formation and melt; changes in water temperatures*

→ concentrate on maximum change between ocean states during the HE compared to the control state | *(1) What is the impact of the duration of the iceberg discharge on the climate's response? (2) To what extent does the simulated signal in d18O calcite during Heinrich event 1 depend on its location? (3) How do the changes in ocean temperatures and d18O seawater caused by the iceberg discharge and related changes in ocean circulation impact the d18O calcite recorded in proxies?*

→ concentrate on complete time series to analyse the signal of d18O_calcite during Heinrich event 1 |
| | |
| **EXPERIMENTAL SET-UP** | |
| Saltfluxes equivalent of 0.2 Sv freshwater | Calving flux (ice) equivalent of 0.2 Sv freshwater that is used to generate icebergs |
| Saltfluxes are added in the North Atlantic | Iceberg calving takes place along the estimated margin of the Laurentide ice sheet |
| Isotopic ratios -20‰ / -30‰ /-40‰ → -20 ‰ fits best to paleoproxy data | Fixed isotopic ratio of -30‰ |
| Duration of 1800 years | Duration of 300 / 600 / 900 years |
| Add artificial addition of salt to the North Atlantic to trigger AMOC recovery | No artificial salt addition. Test how long iLOVECLIM needs to start the AMOC again |
| Compute modelled d18O calcite using the simulated d18O seawater by adding the temperature effect using simulated surface and bottom temperatures (following Shackleton 1974 and Marchitto et al., 2014) | Compute modelled d18O using the modelled d18O seawater and temperatures following the equation of Shackleton:
d18O calcite = 21.9 − 0,27+d18Oseawater (SMOW)-sqrt(310,61-10*Tmodelled) |
| Compute paleoproxy d18O seawater by removing the temperature effect from the paleoproxy-derived d18O calcite | |
| Investigate North Atlantic Deep Water Formation, North Pacific Deep Water formation (restricted impact to North Pacific), Antarctic Bottom Water | Investigate AMOC only |
| Compare 36 cores | Compare 4 cores |
| Determine surface and benthic d18O HS anomalies = difference of 500 year d18O average before and during HE for each core | Display complete time series of cores |
| Compute difference of 10 year averages of control state and end of HE | Display difference of 100 year averages of control state and end of the HE and complete time series of experiments |
| | |
| **RESULTS** | |
| NADW shut down after 200 years | AMOC strongly weakened after 300 years, shutdown after 400 years of iceberg calving |
| Investigate maximum difference between HE state and ctrl state in various depths (10 year averages) | Investigate maximum difference between HE state and ctrl state at the surface (100 year averages) |
| Surface: Maximum change in d18O seawater / calcite, SST in the Iceland Sea and the eastern North Atlantic | Maximum change in d18O seawater / calcite in the Labrador Sea and the central North Atlantic, SST in the central North Atlantic and Iceland Sea |

| Analyse the time development of d18O anomalies (seawater & calcite) in the North Atlantic, North Pacific at top and bottom | Analyse d18O seawater/ calcite / SST signal at different locations to investigate the time development of the d18O_calcite signal during the iceberg calving and the impact of the SST and the d18O_seawater respectively |
|---|---|
| Investigate the impact of the circulation and climate signal, meltwater signal & temperature effect signal | Investigate the impact of SST and d18O_seawater signal on d18O_calcite |

There is however one exciting new aspect in this study: the simulation of more realistic meltwater pattern and the fact that seawater temperature changes due to latent heat loss of melting icebergs are taken into account. To make this study more original, one solution might be to rewrite the paper and focus on the impact of a more realistic parameterization of iceberg drift and melting on d18O seawater and d18O calcite (i.e. compare pure hosing experiments with the same amount of total freshwater with the simulations presented here). Given that most modeling studies of Heinrich events are based on simplified meltwater hosing scenarios, such a study would be of great interest to the modeling community. An in-depth discussion on how well such a coarse resolution model can be expected to capture iceberg drift should also be included.

*As stated above, there are substantial difference between the present study and the paper of Bagniewski et al. (2015), therefore, we do not agree that it is necessary to rewrite the paper. However, we do agree with the good comment of the reviewer that a comparison between a hosing and an iceberg experiment should be included. Thus, we added a paragraph where we compare the hosing experiment – LS 0.2Sv - conducted by Roche et al. (2014) to our ICE-300 experiment, please see general comments.*

*Further, we don't agree that an in-depth discussion on how well such a coarse resolution model can be expected to capture iceberg drift should be included because this issue has already been discussed in detail previous publications (Jongma et al., 2009; Bügelmayer et al., 2015)*

Second, the authors cannot conclude that it is the length of the Heinrich Event that causes the non-linear response of the AMOC. To make this conclusion, one additional simulation should be integrated which introduces the same overall volume of icebergs released in ICE-900 over only 300 years. I would not be surprised if the impact on the AMOC in such a simulation is similar to ICE-900, but I might be wrong. There are many publications discussing the hysteresis behavior of the AMOC for a whole zoo

of models (including model intercomparison studies), and I would strongly encourage the authors to cite the original literature when discussing this aspect. Most of these publications base their analysis on the total volume of freshwater added. If LOVECLIM shows a significantly different behavior based on the duration only (and not on total volume) then this might be an interesting and publishable result. Otherwise, there is nothing new about this result.

*As suggested by the reviewer, we have performed one additional experiment where we added 0.6 Sv over 300 years. The AMOC shuts down immediately when applying such a strong forcing. It increases to about 5 Sv for 300 years and then drops to 4 Sv. In total it takes 800 years in iLOVECLIM for the AMOC to recover from a 0.6 Sv iceberg forcing applied for 300 years.*
*The non-linear behavior of the AMOC in iLOVECLIM can be seen in Figure 1 and also Table 1. First, the same amount of freshwater (3\*ICE-300 and ICE-900) results in very different recovery times (800 vs 2200 years). Second, the recovery time does not increase linearly from ICE-300 to ICE-600 and ICE-900 (immediate recovery, 850 year delay and 2200 year delay). Therefore, we conclude that the recovery time of the AMOC depends strongly, but non-linearly on the duration of the forcing applied.*

[Figure]

Figure 1: AMOC of 0.6Sv experiment (0.6Sv applied for 300 years – pink line); ICE-300: green line; ICE-600: red line; ICE-900: black line

Table 1: Summary of the impact of the duration of the iceberg forcing on the recovery time of the AMOC

| Experiment | Strength of forcing (Sv) | Length of forcing (years) | Sea level rise (m) | Recovery time AMOC (years) |
|---|---|---|---|---|
| 3*ICE-300 | 0,6 | 300 | 15,8 | 800 |
| ICE-300 | 0,2 | 300 | 5,3 | 0 |
| ICE-600 | 0,2 | 600 | 10,5 | 800 |
| ICE-900 | 0,2 | 900 | 15,8 | 2200 |

Third, the comparison with only 4 sediment cores is deceptive. There are more cores in the North Atlantic that recorded d18O calcite changes over this period with high enough sedimentation rates. These cores should be taken into account in this analysis.

*Please see general comments.*

Finally, time series of the model results at the location of these cores should be shown with the sediment data in the same figure (for the better or worse) and discrepancies should be discussed. Nobody would expect a perfect fit, but this one-on-one comparison is still important to validate the model simulations.

*As we clearly stated in the manuscript, we do not show the model – data comparison at the core locations, because first, we can't expect the ocean model with a 3°x3° resolution to explicitly reproduce the correct location of local and regional features, such as frontal systems. In our view, it is more important to compare grid cells that represent these features in the model with appropriate core data. In other words, our study focuses on an improved understanding of the system rather than to validation of the model results.*

Other comments:

Proxy data reconstructions and a few model simulations suggest that the main source of dense waters in the North Atlantic during the last glacial was the Nordic Seas. Labrador Sea Water seems to have played a small (or maybe even absent) role. LOVE-CLIM simulates an important convection site in the Labrador Sea for the control run. During the transient simulations, this convection site shuts off and influences d18O seawater, temperatures and, of course, the strength of the AMOC discussed in this paper. The fact that this convection site might or might not be realistic and how this impacts the results should be discussed in the paper.

*Thank you for pointing this out, you are correct that iLOVECLIM wrongly simulates a convection site in the Labrador Sea, which should be located in the Irminger Sea, during the control state. However, as you also pointed out, it is immediately shut down as soon as the iceberg forcing starts and its effect on the results is much smaller than the impact of the icebergs generated.*
*Moreover, it is important to notice that we didn't perform transient simulations to compute the evolution of HS1, since the fronts are not at the exact right position in iLOVECLIM due to its coarse resolution.*

Along the same lines, wouldn't one expect Baffin Bay to be completely ice covered and at freezing point just before and during Heinrich Event 1? How realistic are the simulated warm conditions in this region (probably related to the near-by convection site) at this point of time? See for example Gibb et al. 2015, "Diachronous evolution of sea surface conditions in the Labrador Sea and Baffin Bay since the last deglaciation", The Holocene 25 (12), 1882-1897

*Please note that in the ctrl state (figure 4) the Baffin Bay is at 0°C and ice covered, the dark blue area in Fig. 2b corresponds to 0°-3°C and starts just below Baffin Bay and the white area in Fig. 4e corresponds to ice free conditions.*

What is the equivalent sea level rise for each of the 3 simulations and how does this

compare to data/reconstructions?

*We have added the sea level rise in Table 1.*

Which equation is used to calculate d18O calcite in the model?

*The equation of Shackleton (1976) is used and we added this information in the methods section.*

Line 36: "ocean cores" should probably read "sediment cores"

*Thank you for pointing this out, we changed it.*

Line 55: IRD transported by sea ice?

*You are right, it should only state transported by icebergs.*

Line 56: Should read ">63um"?

*Yes, we changed it accordingly.*

Line 158-159: How can the value of iceberg d18O not be important for a study that analyses changes in d18O during a Heinrich event?

*This was badly formulated, we wanted to express that we implement a fixed value of -30‰, it would be more accurate to receive the value from an ice sheet model itself, but for the current study we don't concentrate on the uncertainties related to this assumption.*

Line 187: it is not obvious (Fig 2) that ICE-600 recovers 700 years later

*This is correct, the (not shown) in line 188 corresponds to both, ICE-600 and ICE-900 recovery time, but we added (please see supplement material) in line 187 to clarify that both, the 700 years and 2,200 are not displayed in Figure 2.*

---

## Author Comment (AC2) · 13 Sep 2016

*Dear Reviewer,*

*Thank you for your thoughtful and helpful comments!*
*Please find our answers and changes below.*

Comments on manuscript cp-2016-31 "Disentangling the effect of ocean temperatures and isotopic content on the oxygen - isotope signals in the North Atlantic Ocean during Heinrich Event 1 using a global climate model" by M. Bügelmayer-Blaschek, D. M. Roche, H. Renssen, and C. Waelbroeck

**General comments**

With an Earth system model of intermediate complexity including an iceberg module the authors investigate the distributions of $\delta^{18}O$ of water and of calcite in the North Atlantic ocean during Heinrich event 1. They analyze the temporal evolution of $\delta^{18}O_{calcite}$ and put forward two different geographical patterns: areas where the $\delta^{18}O_{calcite}$ hardly changes (or with large delay) during H1 in contrast to other areas where the $\delta^{18}O_{calcite}$ closely mimics the evolution of that of $\delta^{18}O_{seawater}$.

This is a very interesting research subject which is helpful in the context of improving our understanding of past climates. The method and tools are pertinent. However the analysis of the results is somewhat too qualitative. The draft seems to have been hastily written with several repetitions and inconsistencies. If re-worked thoroughly this could become a very pertinent paper.

Subject to the revisions of the specific comments below I would recommend publication in CP.

**Specific comments**

1. One important aspect which is not addressed in this study is whether the inclusion of an iceberg module in the model does help improving the modeling of $\delta^{18}O$ during Heinrich events or not? In short: is it worth including icebergs in climate models? Does it bring significant improvement of modeling studies? A comparison with the results of already available water hosing experiments performed with the same model would be welcome and significantly add to the value of the present work.

*Please see general comments.*

2. Some of the conclusions are clearly overstated: "The comparison of the model experiments with four marine sediment cores indicates that the experiment with an iceberg forcing of 0.2 Sv for 300 years yields the most reasonable results." (lines 20-21), "we find that the set-up of an iceberg forcing of 0.2 Sv over 300 years yields the most reasonable results" (line 403), and "From the comparison of simulated sea surface temperatures and $\delta^{18}O_{calcite}$ with proxy data, we find the best agreement between model output and data is reached when the iceberg discharge is stopped after 300 years." (lines 475-477). The data-model comparison as performed so far does not provide enough evidence for such conclusions.

3. The (short, lines 381-401) data-model comparison carried on page 12 suffers from several shortcomings.

   1. Cores 2 & 3 (NA87-22 & CH69-K09) are compared to the model results in the Baffin Bay (lines 381-386). Yet, if the map in Fig. 8 is correct, cores 2 & 3 are not expected to be representative of the Baffin Bay, but rather of the North-East Atlantic and the central North Atlantic.

2. ICE-600 et ICE-900 display similar changes in the Nordic Seas as observed in core 3 (ENAM93-21) but this agreement is dismissed on the basis of a lack of agreement for SST which is not represented for that area. Further, this decision somewhat contradicts the (sound) remark that reconstructed SST for high latitudes have high uncertainties (lines 448-451).

*Thank you for those valid points, please see general comments.*

3. We miss the information in panel a) and d) of Fig. 9 to appreciate how the $\delta^{18}O_{calcite}$ from core 4 (KNR316-GPC5) might fit the different experiments.

*We unfortunately truncated the figures, we have changed this.*

4. The duration of the perturbation in cores 1, 2, and 3 appear to be much longer than in the model experiments. How do you explain it?

*It is important to notice that we didn't perform transient experiments, instead we applied a 300 / 600 / 900 year freshwater flux under constant LGM conditions. We first chose to apply a 0.2 Sv forcing over 300 years because Roche et al. (2014) found the best agreement with paleoclimatic data in this set-up, when comparing the maximum change in d18O_calcite during a before a Heinrich event. Yet, the estimated duration of the Heinrich events varies from 250 to 1500 years, therefore we performed three experiments of different time lengths, but we never intended to simulate the transient pattern recorded in data. We repeated that information of fixed boundary conditions in the discussion (390-392).*

The data-model comparison is minimalist and too qualitative. I would recommend the use of additional cores – many are available – to improve that aspect. Wouldn't it also possible to perform cross-correlation between time series (model versus data; between variables)?

*Please see comment above about the chosen cores.*

4. "the timing of the first response to the iceberg forcing coincides between all the experiments" (abstract, lines 10-12;); "Yet, our results show that the timing of the first response to the iceberg forcing in SST, $\delta^{18}O_{seawater}$ and $\delta^{18}O_{calcite}$ coincides between all the experiments in the various regions within 300 years." (conclusion, lines 472-474): this should not be presented as an outcome of the study; given the experimental setup this behavior is expected.

*Thank you for pointing this out, it is badly formulated. We re-wrote lines (550-553) to:*

*Our results show that the timing of the first response to the iceberg forcing in SST, $\delta^{18}O_{seawater}$ and $\delta^{18}O_{calcite}$ coincides between all the experiments in the various regions and is within 300 years. Applying the iceberg forcing for additional 300 (ICE-600) and 600 (ICE-900) years, respectively, causes a shutdown of the AMOC and more negative values in the North Atlantic.*

5. The discussion on MOC (lines 421-434) should be dropped; it is neither relevant nor needed for the present work.

*We do not agree that it should be completely dropped, but we have added more information concerning other studies (lines 490-500).*

6. Question "(1) what is the impact of the duration of the iceberg discharge on the climate's response?" (line 109) should be reformulated. The consequences of the icebergs discharge on North Atlantic Ocean properties are examined but their climate impact is nowhere discussed.

*Thank you for pointing this out. We have reformulated question 1) to: What is the impact of the duration of the iceberg discharge on the AMOC and the North Atlantic Ocean?*

7. It is nowhere mentioned how $\delta^{18}O_{calcite}$ is computed.

*We have added this information in the methods section.*

8. "In these two regions ICE-600 and ICE-900 show much stronger reductions in SSS than ICE-300 at the end of the iceberg discharge because there the decrease is mainly caused by the advection of the fresh surface waters rather than by the amount of icebergs reaching these areas, which is comparable in all three experiments." (lines 223-226). I do not get the argument. Wouldn't it be simply due to the fact that the cumulative freshwater flux is much larger in ICE-600 and in ICE-900? May be reformulate?

*We have rewritten lines (238-242) to clarify this statement:*

*In these two regions ICE-600 and ICE-900 show much stronger reductions in SSS than ICE-300 at the end of the iceberg discharge. This reduction is mainly caused by the advection of the fresh surface waters, rather than by the amount of icebergs reaching these areas, which is comparable in all three experiments.*

9. ..."we find an immediate response in $\delta^{18}O_{calcite}$ to the iceberg release at the calving sites and in the North Atlantic, but it takes more than 100 years to cause a significant change in regions further away from the calving sites" (lines 240-243). This statement appears to be in contradiction with the foregoing discussion and Fig. 3.

*Thank you for pointing this out, we have re-written the sentence so that it now states (254-257):*
*Moreover, we find an immediate response in $\delta^{18}O_{calcite}$ to the iceberg release at the calving sites and in the North Atlantic. In regions further away from the calving sites it takes up to 100 years for the iceberg discharge to cause a significant change.*

**Other comments**

1. line 73: remove "yearly" (brings confusion with Sv units used for fluxes)

   *We changed line 73 as suggested by the reviewer.*

2. line 81: "simulated" is missing an "l"

   *Thank you, we added the missing letter.*

3. lines 87-90: "but the authors notice that the total volume released is similar to the one obtained by Roche et al. (2004)." Based on the numbers given here, the Roche et al. (2004) freshwater volume (0.29 Sv during 250 yr) is nearly 4 times that of Roberts et al. (2014) (0.04 Sv during 500yr). Can you develop the similarity?

*Roberts et al. (2014) state in their table 1 a total ice volume flux of $60x10^4$ $km^3$, Roche et al. (2004) released $85 x10^4$ $km^3$. We changed line 95-96 so that it now states*

*Their set-up indicates a much weaker freshwater flux of 0.04 Sv over 500 years than expressed by previous studies, but the authors notice that the total ice volume released is similar to the one*

*obtained by Roche et al. (2004).*

4. line 95: replace "the take up" by "the uptake"
   *We replaced it.*

5. lines 104-105: the last part of the sentence "who showed that the freshwater flux that yields model results in best agreement with available proxy data evidence is 0.2 Sv" is part of the preceding discussion (lines 70-90) and should be moved there (line 79?)

   *We deleted this part of the sentence.*

6. lines 120-121: suggestion "The  atmospheric model ECBilt (Opsteegh et al., 1998) is a quasi-geostrophic, spectral model  running with a time step of 4 hours on a horizontal T21 truncation..."

7. lines 123-124: "...precipitation is only computed in the  lowest (tropospheric) layer..."

8. line 130: suggested changes " CLIO has a resolution of 3°x3°  with 20 unevenly spaced vertical levels in the ocean"

9. lines 135-135: "The vegetation model  VECODE (Brovkin et al., 1997)  accounts for..."

   *We took the kind advise of the reviewer (points 6 – 9) into account and changed the manuscript accordingly.*

10. lines 137-138: suggested changes " VECODE is forced by temperature , precipitation and $CO_2$ provided by ECBilt and accounts for long-term..."

    *In the used set-up $CO_2$ was not provided by ECBilt.*

11. lines 152-153: suggestion "and their meltwater fluxes are put into the ocean surface layer of the current  water column."

    *We changed it as nicely suggested.*

12. lines 158 & 399: "‰ ", missing space after the per-mil symbol

    *We added the missing space, thank you.*

13. line 164: what does "This value" refer to? Here there should be reference to the work of Kageyama et al. (2010).

    *This value corresponds to 0.2 Sv mentioned in the sentence before. Unfortunately, we don't understand the comment of the reviewer why there should be a reference to Kageyama et al. (2010).*

14. line 187: the MOC recovery for exp ICE-600 is not seen in Fig. 2
    *We added "(please see supplement material)".*

15. lines 188-190: Previous works addressed the issue of MOC recovery in LOVECLIM; e.g., Rahmstrof et al. (2005), Menviel et al. (2008), Kageyama et al. (2010).

*Thank you for pointing this out, we added the work of Menviel et al. (2008) in the discussion.*

16. line 280: suggest " uptake"

17. line 282 suggested "freezing point at  about -2°C"

*We changed the lines as suggested.*

18. lines 350-352: suggested changes: "Before looking in detail at the four cores to investigate whether or not the simulated patterns can be confirmed by the data, several important remarks need to be made.  First, while the sea level rise due to the released icebergs during HS1 is accounted for in our experiments, we do not simulate the background sea level rise starting at 19 ka after the onset of the LGM (Lambeck..."

*We re-wrote it as commented by the reviewer.*

19. lines 395, 399 & 436: ($\sim$ 1Sv)

20. Figures 4 to 7: should specify that $\delta^{18}O_{calcite}$ and $\delta^{18}O_{seawater}$ represent surface values

21. Figures 4 to 7: scales are not legible

22. Figure 9: the left panels are truncated.
23. Figure 9, legend, 3d line: $\delta^{18}O_{sw}$ and $\delta^{18}O_c$ ("c" an "sw" should appear as subscripts).

24. Figure 10: left panels are truncated.

25. Figure 10: unit of IRD in panel a) number of grains/g sediment?

*Thank you for your comments 19 to 25, we changed the manuscript accordingly.*

**References**
- Kageyama, M., A. Paul, D. M. Roche, and C. J. Van Meerbeeck (2010), Modelling glacial climatic millennial-scale variability related to changes in the Atlantic meridional overturning circulation: a review. Quaternary Science Reviews, 29, 2931-2956, doi:10.1016/j.quascirev.2010.05.029.
- Menviel, L., A. Timmermann, A. Mouchet, and O. Timm (2008), Meridional reorganizations of marine and terrestrial productivity during Heinrich events, Paleoceanography, 23, PA1203, doi:10.1029/2007PA001445.
- Rahmstorf, S., M. Crucifix, A. Ganopolski, H. Goosse, I. Kamenkovich, R. Knutti, G. Lohmann, B. Marsh, L. Mysak, and Z. Wang, A. Weaver (2005), Thermohaline circulation hysteresis: a model intercomparison. Geophysical Research Letters, 32, L23605, doi:10.1029/2005GL23655.
- Roche, D., D. Paillard, and E. Cortijo (2004), Constraints on the duration and freshwater release of Heinrich event 4 through isotope modelling, Nature, 432, 379–382, doi:10.1038/nature03059.
- Roche, D. M., D. Paillard, T. Caley, and C. Waelbroeck (2014), LGM hosing approach to Heinrich Event 1: results and perspectives from data–model integration using water isotopes, Quaternary Science Reviews, 106, 247-261, doi:10.1016/j.quascirev.2014.07.020.

---

## Author Comment (AC3) · 13 Sep 2016

**Disentangling the effect of ocean temperatures and isotopic content on the oxygen - isotope signals in the North Atlantic Ocean during Heinrich Event 1 using a global climate model: a review**

This study exmaines how the simulated delta 18O of calcite varies in a climate model when different time durations of Heinrich Events are simulated. Its results are interesting and present a novel way to think about the signal that is in the available data. It was let down, however, by its poor figures and rather superficial analysis.

I had to work really hard to relate what was going on in the text with what was shown in the figures. Many of the figures (e.g. fig. 10) don't really support the claims that are made in the text: the claims may be true, but I can't satisfy myself that they are from the figures. There are more detailed crticisms later but I urge the authors to think when preparing their figures: how can I make it as easy as possible for a reader to understand my figures. At present I feel that the figures have been prepared with what is easiest for the authors in mind. If your figures are hard to decipher readers won't bother to read the paper, so it really is in your interests to think about this.

There are a number of times in the text (for example the last line of the m/s) where claims are made that the simulated calcite is a "reasonable" fit or other such vague phrases. Simple statistics do exist that can quantitatively define how well series relate to one another. These should be used rather than qualitative claims of "reasonableness". That palaeoclimatology is rife with such vagueness is no excuse for this.

*Dear reviewer,*
*Thank you for taking the time to thoroughly comment on our paper. We have changed the figures so that they are now hopefully easier to follow and better support the text. Moreover, we have revised the model - data comparison substantially, as mentioned in the general comments.*

**Specific Comments/questions:**

Motivation:
It would be good to see the motivation for why it was chosen to test the sensitivity of the model to the duration of the simulated Heinrich Events rather than the size. Why vary the duration of the events and not the size? The size (that is flux of ice) and duration are to a certain extent constrained by the total volume of ice that can be released from the Laurentide Ice Sheet, therefore would not a better test to have been to keep the total ice volume constant and thus for the longer duration events have a small flux of ice?

*We focused on the impact of the duration of the Heinrich Event because the impact of the amount of freshwater released over a certain amount of time has already been tested by Roche et al., 2014 using the same model. In their study the authors applied the various freshwater fluxes (e.g. 0.12, 0.18, 0.2) for 300 years and found that a freshwater forcing of 0.2Sv yields the best agreement with data. Therefore, we chose to apply 0.2Sv, but in the current study in the form of icebergs and to investigate the impact of the duration.*

Model description:
Unless I have missed it there is no description of how delta 18O calcite is calculated within the model. This needs to be included along with discussion on how this model quantity relates to the calcite that you might find in forams. For example, is calcite calcuated in a water depth in which *bulloides* lives?

*We have added a short description of the computation of $\delta^{18}O_{calcite}$.*
*In our results we display the calcite at the ocean surface, but we tested the impact of the different ocean depths (surface, 45m, and 90m depths) on the results and found that they were very small.*

Section 4:
Section 4 could be a very nice section but is currently weak. More effort to link the observations back to the model runs would strengthen this. You are, I think, hindered by the currently horrible figure 10 in doing this.

It would be helpful to make some comments on how well you feel your model could in a perfect world simulate the calcite in forams (see above).

*Please see general comments.*

Line 402: "Overall, we see some similarities between the simulated and measured δ 18 Ocalcite curves and we find that the set-up of an iceberg forcing of 0.2 Sv over 300 years yields the most reasonable results compared to the proxy data considered"
If I am honest I do not think that I can judge this because comparing the simulated and proxy calcite is near impossible. See my comments on fig. 10.
The phrase "most reasonable" is unacceptably vague.

The regional abbreviations e.g. stNA, neNA, BB etc. are not necessary and detract considerably from the clarity of the manuscript. Please just use the full description.

*We have deleted these abbreviations from the text, but kept them in Figure 7f.*

Other comments

line 475: "From the comparison of simulated sea surface temperatures and δ 18 Ocalcite with proxy data, we find the best agreement between model output and data is reached when the iceberg discharge is stopped after 300 years."
I'm not sure I know what figure/metric shows this.

*Please see revised discussion.*

**Figure comments.**

Fig 1. Are the calving locations really those weird shapes? Why not use a pseudocolour plot rather than a contouring one to show the location, then one will be able to see what the model grid resolution is.

*Yes, this is the model resolution.*

Figs 5/6/7. These figures should be split up so that all of the panels (a) are on one figure all the (b)s on another etc. In the text the comparisons are always framed in terms of the variable (e.g SST) not the model run (e.g. ICE 600). At present you have to flip between 3 figures for each variable. This is really difficult to do. You shold also make the panels larger: the numbering on the colour scale is invisible.

*We have changed the figures as suggested.*

Fig 8 please put the names of the core on the map.

*We have added the names.*

Fig 9/10. In figure 9 time goes from left to right, in 10 it goes from right to left. So when trying to relate the modelled delta 18O calcite to the proxy delta 18O, not only do you have to flip between figures but you also have to mentally flip them through 180 degrees. Please chose one direction for time and stick to it.

*Yes, this was an unfortunate mistake, thank you for pointing it out.*

The panels in fig 9 are tiny. On full zoom my nose is about 2cm from the screen when I look at them!! Please make them bigger. The axes are also missing from the leftmost panels.

*We have changed figure 9 – now figure 7 so that it is easier readable.*

You must add the simulated calctie 18O curves into fig. 10 so that it is easier to follow the text. Again I have to flip between figure to work out what is happening: this is really hard.

*We decided against adding the simulated d18Ocalcite to the data because we think it will make the figure even more complicated. The revised figure 7 and figure 8 allow for an easier comparison than in the previous version.*

Fig 10. The core that is in panel (a) is the core that is numbered 2 in fig.8. Panel (b) has core one. This is random. Please have panel (a) as core 1, panel (b) core 2 etc. Also as well as the core name put the number in the title.

*Thank you for pointing it out, we have changed it.*

To correctly interpret fig 10, I have to:
    (1) Look at fig8 to work out the number/location of the core,
    (2) read the caption to work out what the core name is.
    (3) look at fig 9 to find the modelled calcite
    (4) look at the title of the plot to find the region
    (5) refer back to fig 8 caption because the region names on the fig8 map and in fig9 are different
    (6) mentally flip the time axis of the panel in Fig.9
    (7) refer back to caption of fig. 10 because I have forgotten which core was which......

This is **7 steps** before I can even look at what the data says.

On the axis in Fig 10 please change the axis label txt to be the same colour as the line to which it refers.

*We have changed figure 10, now figure 8, as kindly suggested by the reviewer. Moreover, we have added the map with the regions and the core locations to the model results so that now only two figures have to be compared, instead of three.*

---

## Author Comment (AC4) · 13 Sep 2016

I attached the answer to your comments as well as the general answers to concerns raised by all of you.

Kind regards Marianne Bügelmayer-Blaschek

Please also note the supplement to this comment:

http://www.clim-past-discuss.net/cp-2016-31/cp-2016-31-AC4-supplement.zip